# Effectiveness of Whole-Body Vibration Training to Improve Muscle Strength and Physical Performance in Older Adults: Prospective, Single-Blinded, Randomized Controlled Trial

**DOI:** 10.3390/healthcare9060652

**Published:** 2021-05-31

**Authors:** Nam-Gyu Jo, Seung-Rok Kang, Myoung-Hwan Ko, Ju-Yul Yoon, Hye-Seong Kim, Kap-Soo Han, Gi-Wook Kim

**Affiliations:** 1Department of Physical Medicine & Rehabilitation, Jeonbuk National University Medical School, Jeonju 54907, Korea; cnk9016@jbnu.ac.kr (N.-G.J.); mhko@jbnu.ac.kr (M.-H.K.); spcmoon@gmail.com (J.-Y.Y.); 2Research Institute of Clinical Medicine of Jeonbuk National University-Biomedical Research Institute of Jeonbuk National University Hospital, Jeonju 54907, Korea; srkang@mdctc.or.kr (S.-R.K.); hskim@mdctc.or.kr (H.-S.K.)

**Keywords:** physical exercise, muscle strength, physical activity, quality of life, aged

## Abstract

Whole-body vibration training (WBVT) is emerging as an alternative exercise method that be easily performed by older adults. This clinical trial investigates the efficacy of WBVT in improving muscle strength and physical performance before resistance exercise, in comparison to conventional resistance exercise after stretching exercise in older adults. The WBVT group (*n* = 20) performed WBVT using a vibrating platform (SW-VC15™), followed by strengthening exercises. The control group (*n* = 20) performed stretching instead of WBVT. Both groups underwent a total of 12 sessions (50 min per session). The primary outcome was isokinetic dynamometer. The secondary outcomes were grip strength, short physical performance battery (SPPB), a 36-Item Short Form Survey (SF-36), and body composition analysis. In all results, only the time effect was significant, and the group effect or time x group effect was not. Both groups showed a significant increase in isokinetic dynamometer. Although there was no significant group effect, the increase in mean peak torque was greater in the WBVT group. The only WBVT group showed significant improvement in SPPB. In SF-36, only the control group showed significant improvements. WBVT can be safely performed by older adults and may be an alternative exercise method to boost the effect of strengthening exercise.

## 1. Introduction

The main physical changes from aging include hormonal changes, decreased neuroregulation, and muscle mass loss. Sarcopenia is defined as the loss of muscle mass and strength with aging [1]. Muscle weakness and decreased strength production are major factors in reducing functional activity. Sarcopenia also limits daily activity and increases the risk of falls. Falls may lead to fractures, which in turn reduce quality of life, increase fatalities, and increase social costs for healthcare [2].

For this reason, the importance of improving muscle function in the older adults is being increasingly emphasized [3]. This usually involves various forms of strength training, including lifting free weights, using resistance bands or body weight for resistance, weight machines, or other methods [4]. Especially for older adults, easy and safe exercises such as squats, lunges, or shoulder presses are recommended [5]. For older people, however, conventional exercises may be difficult to perform due to the deterioration of their physical capacities [6].

Whole-body vibration training (WBVT) is emerging as an alternative way of improving neuromuscular function in the physical therapy field [2]. In general, WBVT can be easily applied to older adult patients either at home or at rehabilitation centers because trainees are only required to stand on a vibrating platform during WBVT [7]. WBVT has also been shown to have positive effects on all organs of the human body, such as improving strength, balance, functional mobility, quality of life, reducing oxygen intake, as well as increasing blood flow and body fat percentage, bone mineral density, cardiopulmonary function, and vascular function in the older adults [8,9,10]. In addition, WBVT has been shown to be effective in a wide variety of fields, such as stroke and cerebral palsy rehabilitation, musculoskeletal pain, and blood sugar control effect in diabetic patients [11,12,13,14,15]. A recent experimental study demonstrated that WBVT improved the physical ability of the older adults with sarcopenia [16]. Some studies have suggested that systemic vibration is more effective in the rehabilitation of the older adults and patients than in the general public [17].

Based on the evidence that WBVT is effective in muscle strength [9], up to now, many studies have compared WBVT and resistance exercise, or compared the simultaneous resistance exercise with WBVT and WBVT alone. However, there is insufficient research or evidence of how much WBVT will boost the effect of resistance exercise compared to conventional resistance exercise after stretching exercise [18,19,20,21,22,23]. In addition, the effect on quality of life after WBVT is still unclear [9]. The purpose of this study is to verify the efficacy and safety of WBVT in improving muscle strength and physical performance before resistance exercise in older adults, in comparison to conventional resistance exercise after stretching exercise.

## 2. Materials and Methods

### 2.1. Study Design

This study was a single-center, prospective, single-blinded (assessor), randomized controlled trial, conducted at the Rehabilitation Center of Jeonbuk National University Hospital from April 2019 to October 2019. Each participant who fulfilled the inclusion criteria was randomly assigned to one of two groups. A clinical research coordinator who was a clinical research nurse and not involved in the assessments was responsible for assigning participants. Group assignments were thus concealed from all of the investigators. All participants were assessed by two investigators who were blinded to group assignments.

Study participants were randomized into 1 of 2 groups and underwent different training protocols. The WBVT group first performed WBVT (20 min) using a vibrating platform (SW-VC15™, SONICWORLD Co., Ltd., Wonju, Korea), followed by strengthening exercises (20 min) after a 10-min break. The control group simply performed stretching (20 min) instead of undergoing WBVT. Stretching before exercise has a beneficial effect on injury prevention [24]. Both groups underwent a 50-min training regime at each session, which was performed for a total of 12 training sessions (3 sessions weekly for 4 weeks). Measurements were conducted at three time points: evaluation 1 (E1, pre-treatment), evaluation 2 (E2, post-treatment), and evaluation 3 (E3, 4 weeks after treatment).

The study is registered at the Clinical Research Information Service, under the direction of the Korea Centers for Disease Control and Prevention (Registration number: KCT0004831).

### 2.2. Randomization

Study participants were randomized into either the WBVT group or the control group and underwent different intervention protocols. Simple randomization was done using the random number table of the Randomization.com website. Seed values used at randomization were also recorded. Randomization was completed before the first participant screening, and results were generated by a researcher not involved in study enrollment or intervention. The assessors collecting study data were unaware of group assignments throughout the trial.

### 2.3. Participants

This study was approved by the Institutional Review Board of Jeonbuk National University Hospital (Approval number: CUH 2018-07-035-003). Study participants were recruited through a notice posted on the bulletin board of the hospital and screened by a rehabilitation physician. Written informed consent was obtained from all participants before randomization. All research procedures were conducted in accordance with the ethical standards of the Declaration of Helsinki.

Patients were selected based on inclusion criteria. First, they were required to be over 65 years of age. Second, they were required not to experience dizziness as a result of vibration. Third, they were required to have no communication problems when conducting clinical trials (K-MMSE, Korean-Mini Mental Status Examinatio n≥23) [25]. Fourth, they were required to fully understand the purpose and procedures of the research and confirm their willingness to participate in clinical research through voluntary consent. Exclusion criteria included: First, subjects who had experienced musculoskeletal injury within the preceding 6 months. Second, subjects with cardiovascular disease. Third, subjects deemed to be ineligible for participation after examination by the research director.

### 2.4. Sample Size

As an investigator-initiated trial to verify the efficacy and safety of muscle strength and physical performance using the equipment (vibrating platform, SW-VC15^TM^) in participants over 65, Ekaterina Tankisheva et al. studied the improvement of lower extremity muscle function evaluated by isokinetic dynamometer in the stroke patients using Power Plate^®^ (Power Plate Acquisitions, LLC, 160 Jan van Gentstraat, 1171GP, Badhoevedorp, The Netherlands), which is similar to our trial device [26]. The expected average difference between the whole body vibration group (*n* = 7) and the control group (*n* = 6) was 21.4, and the standard deviation was 18.069. The sample size was calculated as 14, considering the drop rate of 20% according to the calculation formula. In this study, 40 patients (20 per each group) were assigned to each group.

### 2.5. Intervention

WBVT was conducted using a 900 × 910 × 1450 mm-sized vibrating platform (SW-VC15™, SONICWORLD Co., Ltd., Wonju, Korea) (Figure 1) [27]. This device transmits sound wave vibration to the human body by generating precise vertical vibration, not horizontal vibration, as in previous types of mechanical acceleration [28]. The device enables the appropriate frequency (3–50 Hz) and amplitude (0–99 mm) to be set according to a patient’s condition. In this case, vibration frequency was set to 10 Hz and vibration amplitude was set to 5 mm [29].

The WBVT group performed WBVT (20 min) using the vibrating platform (SW-VC15™), followed by strengthening exercise (20 min) after a 10-min break. During WBVT, participants were instructed to stand upright on the platform. Strengthening exercises were body weight exercises without additional weight loading, and included rowing exercises (flexing the arms until hands were aligned with the chest, and stretching arms while bending forward 45° from the waist), body weight squat exercises (flexing thighs and knees until thighs became parallel to the floor and fully extended), front lunge exercises (stepping forward with one foot, flexing legs until the rear calf became parallel to the ground with leg extended), and shoulder exercises (flexing arms parallel to the ground with arms flat). Each exercise was performed slowly over 4 s per repetition, and all exercises were performed equally for 20 min [30,31,32,33,34].

The control group performed stretching only (20 min) instead of WBVT, followed by the strengthening exercises (20 min) after a 10-min break. The stretching was a slow movement exercise, and the following 10 movements were prepared: 1. Neck circumduction; 2. Neck flexion and extension; 3. Neck bending side to side once; 4. Shoulder circumduction; 5. Hip circle exercise; 6. Pendulum leg swing front to back; 7. Pendulum leg swing side to side; 8. Raise knees higher and higher with each step for hamstring stretching; 9. Pull heel in closer toward buttock with each step for quadriceps stretching; 10. Ankle rotation stretching. Each motion was performed slowly over 4 s per repetition to the point of mild discomfort. Subjects performed each motion for 30 s and then followed with the next motion. It took a total of 5 min to perform all 10 stretches for 30 s, and this was repeated 4 times for 20 min [30,35].

WBVT was conducted individually, and stretching and strengthening exercise were conducted by group training in a physical therapy room. All training courses were guided and supervised by a clinical research nurse to ensure that all participants faithfully exercised.

### 2.6. Outcome Measurements

Two-blinded evaluators who did not participate in the intervention performed all measurements. For one subject, one evaluator was in charge of three evaluations.

#### 2.6.1. Primary Outcome Measurements

The primary outcome was the isokinetic dynamometer. Knee flexor and extensor muscle strengths were measured with an isokinetic dynamometer (Model 900-240; Biodex Medical Systems, Shirley, NY, USA). The isokinetic dynamometer evaluates peak torque (N·m) and average power (Watts) at 60 deg/sec during flexion and extension of bilateral knee joints in sitting position [36]. Peak torque means the highest muscular force output at any moment during a repetition and is indicative of a muscle’s strength capabilities. Average power means total work divided by time and represents how quickly a muscle can produce force [37]. The mean peak torque and average power were obtained by averaging the values of these 4 movements (flexion and extension of bilateral knee joints).

#### 2.6.2. Secondary Outcome Measurements

The secondary outcomes were hand grip strength, short physical performance battery (SPPB), a 36-Item Short Form Survey (SF-36), and body composition analysis.

Hand grip strength was also used to measure muscle strength. For grip strength, a hand dynamometer (JAMAR^®^, Chicago, IL, USA) was used to measure maximum bilateral hand strength (kg). A high value was obtained by measuring bilateral hand strength twice in each case, from which mean values were calculated [38].

SPPB was evaluated to check the balance function and was based on three components: balance, gait speed, and chair stand tasks (0–4 points per item, total scoring from 0 to 12). The balance component consisted of three sub-components: side-by-side standing (1 pt), semi-tandem standing (1 pt), and full tandem standing (2 pts). For the balance tasks, participants were asked to stand with their feet side-by-side, followed by the semi-tandem (heel of one foot alongside big toe of other foot) and tandem (heel of one foot directly in front of and touching other foot) positions for 10 s each. For gait speed, a 4-m walk at the participant’s usual pace was timed. The test was conducted twice, with the faster of the two walks used. For repetitions of the chair stand task, participants were asked to stand up and sit down five times as quickly as possible with arms folded across their chests. This test was performed only after participants had first demonstrated the ability to rise once without using their arms. Higher scores indicated good physical function [39].

The 36-Item Short Form Survey (SF-36) was a self-administered questionnaire comprising 36 items that surveyed overall health status. It measured health across eight multi-item dimensions, covering physical functioning, physical role limitations, emotional role limitations, energy/vitality, mental health, social functioning, pain, and perception of general health. Pre-coded numeric values were recoded according to a score from 0 to 100. That is, each item was scored on a 0–100 range. For a given dimension, scores of components were averaged together. The percentage scores across all of the the eight dimensions were summed and divided by 8 to arrive at a global score, with a higher score indicating a better health state [40].

Body composition analysis was performed to evaluate body weight, skeletal muscle mass, and body fat mass using an InBody^®^720 Portable Body Composition Analyser (Biospace co., Ltd., Seoul, Korea). The InBody^®^720 analyzes body composition by analyzing the impedance of body tissue. Measurements were taken in the morning to minimize the effects of diurnal fluctuations [41].

#### 2.6.3. Adverse Effects

Safety was assessed by monitoring adverse reactions, such as subjective awareness or symptoms, measurement of vital signs, and self-reporting by participants.

### 2.7. Statistical Analysis

Statistical analysis was performed using SPSS 23.0 software for Windows (SSPS Inc., Chicago, IL, USA). Data were presented as mean (SD) for continuous variables and frequency for categorical variables. For baseline difference (other than sex), an independent *t*-test was used when the assumption of normality was satisfied; otherwise, the Mann–Whitney U-test was used. The Pearson Chi-square test was used to compare differences between groups for categorical variables by demographic (sex). The outcome data were analyzed using repeated measures (RM) analysis of variance (ANOVA). When the time effect was significant, intra-group analysis was performed through additional RM-ANOVA. Each RM-ANOVA was followed by planned multiple pairwise comparisons with Bonferroni correction to *p* ≤ 0.05.

## 3. Results

### 3.1. Participants

Forty participants, 20 in the control group and 20 in the WBVT group, were recruited between 09 April 2019 and 04 October 2019. Twenty participants were randomly allocated to each group. All participants completed the treatment and evaluation (baseline, post-treatment and 4-week follow-up evaluation) (Figure 2).

Demographic data and baseline characteristics for each group are shown in Table 1. There were no significant differences in baseline demographic between the two groups.

### 3.2. Primary Outcomes

The RM-ANOVA analysis for mean peak torque revealed significant main effect for Time (F = 13.781, *p* < 0.001), but no Group effect (F = 0.162, *p* = 0.690) and Time × Group effect (F = 1.645, *p* = 0.200). Analysis for average power revealed significant main effect for Time (F = 20.511, *p* < 0.001), but no Group effect (F = 0.682, *p* = 0.414) and Time × Group effect (F = 0.319, *p* = 0.675) (Table 2).

In the intra-group comparison, both groups showed a significant increase in mean peak torque (RM-AVOVA, *p* = 0.002 in WBVT and *p* = 0.003 in control) and mean average power torque (RM-AVOVA, *p* = 0.004 in WBVT and *p* < 0.001 in control). Planned pairwise comparisons revealed significant differences in E1 versus E2 (*p* = 0.009 in WBVT and *p* = 0.048 in control) and E1 versus E3 (*p* = 0.045 in WBVT and *p* = 0.012 in control) of mean peak torque in both groups, and also in E1 versus E2 (*p* = 0.009 in WBVT and *p* = 0.003 in control) and E1 versus E3 (*p* = 0.033 in WBVT and *p* = 0.001 in control) of mean average power in both groups (Table 3).

Although there was no significant Group effect, the increase in mean peak torque was greater in the WBVT group than in the control group. (ΔE2-E1: 7.03 ± 9.26 versus 3.49 ± 5.89).

### 3.3. Secondary Outcomes

#### 3.3.1. Grip Strength

The RM-ANOVA analysis for mean grip strength revealed significant main effect for Time (F = 3.626, *p* = 0.038), but no Group effect (F = 0.004, *p* = 0.948) and Time × Group effect (F = 1.431, *p* = 0.246) (Table 4). Although both groups showed slightly increased grip strength after treatment, and the main effect of Time was significant, there were no significant intra-group comparison results in both groups, respectively. (Table 5).

#### 3.3.2. Short Physical Performance Battery

The results of RM-ANOVA for SPPB revealed significant main effect for Time (F = 6.375, *p* = 0.003), but no Group effect (F = 1.073, *p* = 0.307) and Time × Group effect (F = 0.536, *p* = 0.587) (Table 4). In intra-group comparison, the only WBVT group showed overall significant improvement (*p* = 0.014), and planned pairwise comparisons revealed significant differences between E1 versus E2 (*p* = 0.025), but no E1 versus E2 (*p* = 0.058) and E2 versus E3 (*p* = 1.000). In the control group, although the score of SPPB was increased, there was no overall significant difference (*p* = 0.104) (Table 5).

#### 3.3.3. 36-Item Short Form Survey

The results of RM-ANOVA for SF-36 revealed significant main effect for Time (F = 5.483, *p* = 0.006), but no Group effect (F = 2.075, *p* = 0.052) and Time × Group effect (F = 3.104, *p* = 0.051) (Table 4) In intra-group comparison. The only control group showed overall significant improvement (*p* < 0.001) and planned pairwise comparisons revealed significant differences between E1 versus E2 (*p* = 0.004) and E1 versus E3 (*p* = 0.004), but no E2 versus E3 (*p* = 1.000). In the WBVT group, although the score of SF-36 was slightly increased, there was no overall significant difference (*p* = 0.771) (Table 5).

#### 3.3.4. Body Composition Analysis

In the results of RM-ANOVA for body composition analysis, only the main effect for Time of skeletal muscle mass was significant (F = 4.611, *p* = 0.017), other results were not significant (Table 4).

In the intra-group comparison, the only control group showed a overall significant decrease in skeletal muscle mass (RM-AVOVA, *p* = 0.233 in WBVT and *p* = 0.022 in control). Planned pairwise comparisons revealed a significant difference in E1 versus E2 in skeletal muscle mass of the control group (*p* = 0.007), but no differences in E1 versus E3 (*p* = 0.233) and E2 versus E3 (*p* = 1.000) (Table 5).

No clinically significant adverse events were observed during the study.

## 4. Discussion

This study showed a significant increase in lower-extremity muscle strength in both the WBVT and control groups. Although there was no significant Group effect, the increase in mean peak torque was greater in the WBVT group than in the control group. In the SPPB score, only the WBVT group showed a significant increase. Only the control group showed a significant decrease in skeletal muscle mass in the body composition analysis, and a significant increase in SF-36. There was no significant difference between the two groups.

Decreased muscle mass, poor balance function, and physical performance are major social concerns [42]. Hormonal changes, decreased bone density and risk of falling are well-known concerns in older adults. In particular, sarcopenia and muscle wasting are closely related to physical performance and mortality [43]. There is currently active research on the nutritional and practical management of sarcopenia. In older adults, exercise, such as walking, muscle strengthening, and stretching, can improve physical activity, quality of life, and reduce psychological distress [44]. Exercise also plays an effective role in cognitive function in the older adults [45]. In older patients, physical performance was also improved through home-based or community-based exercise therapy [46].

Resistance exercise is effective in increasing muscle strength, while stretching is effective for muscle function and fall prevention for the older adults [31]. In the present study, both groups performed resistance exercise, while the control group only performed stretching exercise. Based on the above studies, we selected resistance and stretching exercises appropriate for older adult participants and produced videos of exercise methods. Participants gathered in groups and followed the exercise instructions explained in the videos under the supervision of the clinical research nurse.

Previous studies reported that acute static stretching decreased muscle strength [47]. This negative effect is known to be especially affected when stretching is performed immediately prior to the muscle strength activity and when doing activities that require high power [24,48]. However, in our study, it was slow movement exercise and performed with relatively low intensity (point of mild discomfort), and strengthening exercise (body weight exercise, not strenuous exercise) was performed after 10 min of rest. In our protocol, the decrease in muscle strength by stretching was considered to be minimal.

Depending on the progression of disability in the older adults, there may be limitations in their ability to perform exercise. The wider context for the popularity of WBVT is that it is easy to apply, even in older adults with poor activity ability [49]. The vibration produced by the vibration platform is transmitted through sensory receptors. The transmission of this mechanical vibration triggers a tonic vibration reflex (TVR), a complex spinal and supraspinal neurophysiological reaction. This TVR can in turn activate muscles and improve physical performance [50]. When stimulated by vibration, tactile feedback and intrinsic sensations help to control posture stability [51].

In the present study, both groups performed strengthening exercises and showed a significant increase in lower-extremity muscle strength. Although there was no significant Group effect, the increase in mean peak torque was greater in the WBVT group than in the control group. Previous studies have shown that WBVT increases testosterone, growth hormone, and IGF-1 levels, resulting in increased muscle strength [52]. Some previous studies have found that WBVT had similar effects compared to general fitness [49]. Our findings were thus similar to those of these studies.

Previous WBVT studies of older adults that have compared an experimental group combining WBVT with resistance exercise and a control group without WBVT for 12 weeks have shown a significant increase both in strength and total SBBP score [53]. In the total SPPB score of our results, only the WBVT group showed significant improvement (E1 versus E2). Our study is thus consistent with the results of previous studies. We additionally conducted statistical analysis of raw data (recording time(s) of the gait speed and chair stand exercises). The control group showed no significant result in the total SPPB score but showed better results than the WBVT group in gait speed and chair stand task. This may be because stretching is particularly effective for flexibility and gait speed, as reported in previous studies [54].

In body composition analysis, neither body weight nor body fat mass showed significant results in either group. Skeletal muscle mass also did not increase in the WBVT group, although it showed a significant decrease in the control group. In previous studies, vibration training has been known to cause tonic vibration reflex, muscle contraction, and consequently an increase in muscle mass, as well as increasing testosterone and IGF-1 and decreasing cortisol, resulting in a decrease in fat mass [49]. In another study comparing vibration training and conventional exercise over 24 weeks, a significant decrease in body fat composition was observed in both groups [55]. For maintaining muscle mass and preventing weight gain, the recommended exercise regimen for the older adults is moderate-intensity aerobic and strengthening exercise for 150 min, twice a week [56]. In our study, it may have been difficult to obtain a significant increase in muscle mass by exercising for only three times a week, over a total of 12 sessions. Further studies with relatively longer timeframes and larger numbers of participants will enable more meaningful analysis of body composition.

Previous studies have shown that WBVT raises serotonin levels in the brain. Serotonin is also known to be linked to quality of life [57]. In another study that applied WBVT for 6 weeks to community-dwelling older adults, there was a significant increase in quality of life (SF-36) [58]. The effect of WBVT on mood has also been shown, although in the participants studied were young athletes, and it has to date not been clearly demonstrated in older adults [59]. Our results for the quality-of-life satisfaction test (SF-36) showed a significant increase only in the control group, after rather than before training. The difference in change (E1 versus E2) in the control group was almost five times higher (ΔE2-E1: 2.10 ± 15.19 versus 10.30 ± 12.11). Although the score for the WBVT group did also increase, the effect of WBVT on quality of life in the present study was not significant. Rather than the static position required for WBVT, perhaps the movements required in performing stretching exercises as a group may have been more satisfying to the participants.

One of the limitations of our study was that groups with various combinations of exercise (WBVT, stretching, and strengthening) and a sham stimulation WBVT group could not be arranged. If so, it would be helpful to know which exercises were more effective in older adults. Another limitation was that the study was able to include only a relatively small number of exercise sessions. The other limitation was that the study did not assess long-term effect and had relatively small sample size. We are planning to address this issue in future studies.

## 5. Conclusions

Lower extremity muscle strength improved significantly in both the WBVT (WBVT + strengthening) and the control (stretching + strengthening) groups, and physical performance improved only in the WBVT group after training. WBVT can be safely performed by older adults and may be an alternative exercise method to boost the effect of strengthening exercise.

## Figures and Tables

**Figure 1 healthcare-09-00652-f001:**
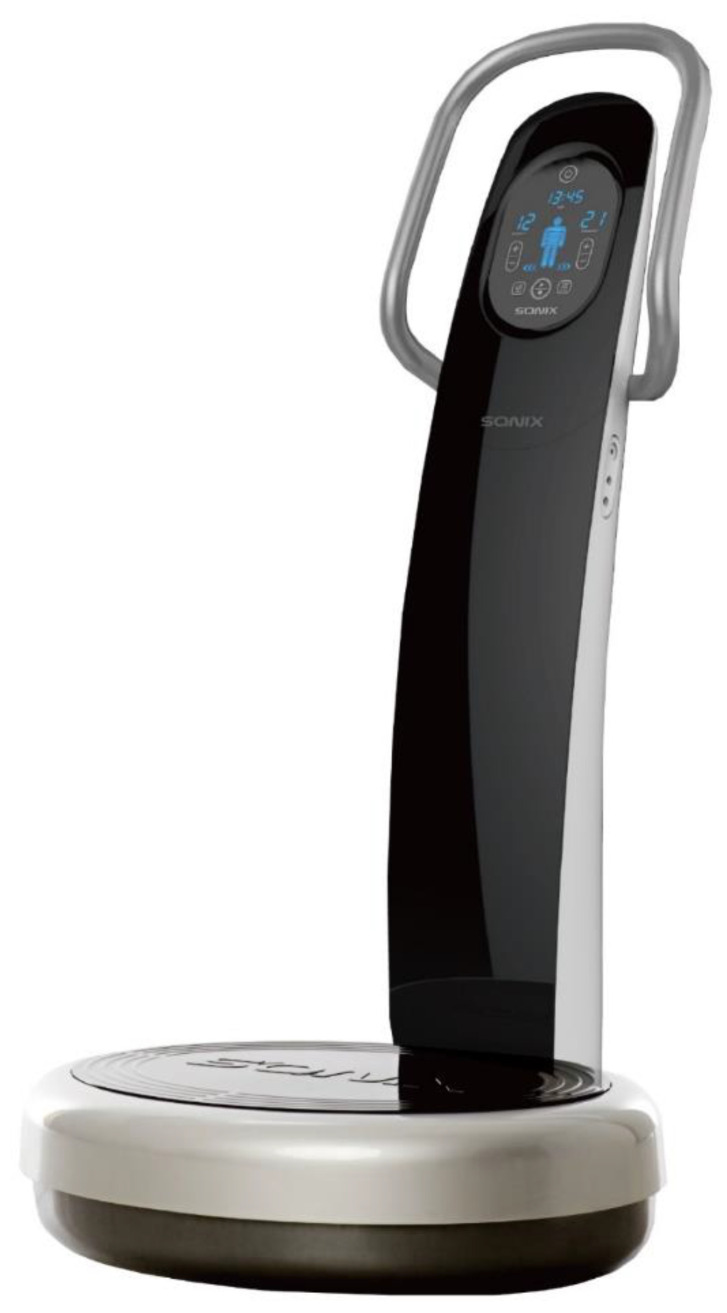
SW-VC15™.

**Figure 2 healthcare-09-00652-f002:**
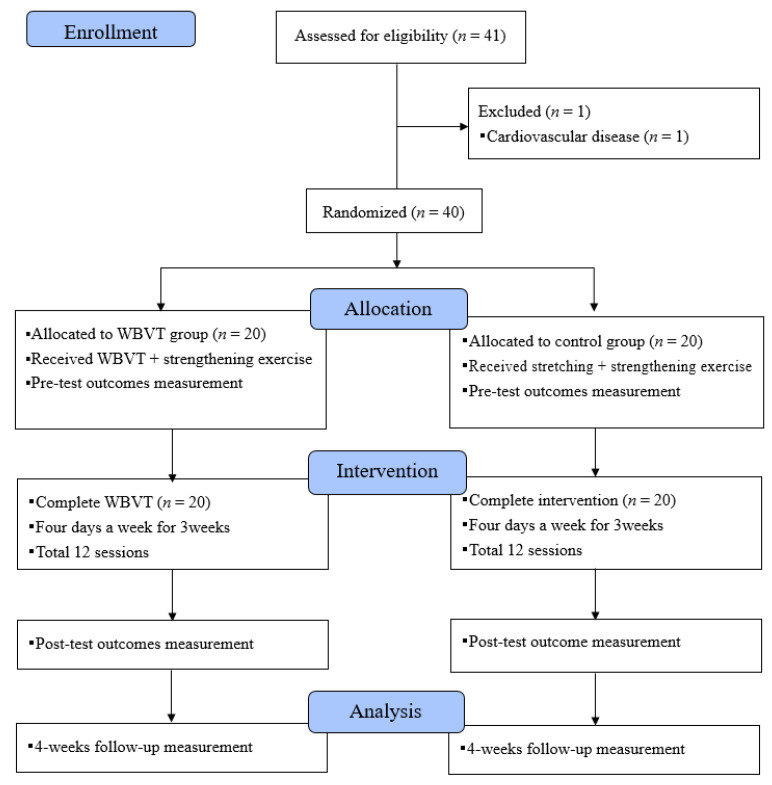
Participant flow diagram. Abbreviations; WPBT, whole-body vibration training.

**Table 1 healthcare-09-00652-t001:** Comparison of baseline demographics and outcome measures between groups.

Variable	Control Group (*N* = 20)	WBVT Group (*N* = 20)	*p*-Value
Age	73.45 ± 4.58	74.35 ± 3.53	0.491 ^1^
Sex (Men/Women)	9/11	9/11	1.000 ^3^
K-MMSE	28.80 ± 1.11	28.30 ± 1.34	0.225 ^2^

Abbreviations: K-MMSE, Korean-Mini Mental Status Examination; WBVT, Whole Body Vibration Training; ^1^ Independent *t* test, ^2^ Mann-Whitney U test, ^3^ Pearson Chi-square.

**Table 2 healthcare-09-00652-t002:** Primary outcome measures (Isokinetic dynamometer).

	E1		E2		E3		Effect Time			Effect Time × Group		
	WBVT	Control	WBVT	Control	WBVT	Control	*F*	*p*	$\eta^{2}p$	*F*	*p*	$\eta^{2}p$
Mean peak torque	48.18 (11.64)	51.61 (22.13)	55.20 (13.88)	55.10 (22.61)	53.38 (14.50)	57.02 (23.90)	13.781	<0.001	0.266	1.645	0.200	0.041
Mean average power	24.43 (5.90)	26.96 (13.31)	28.18 (7.08)	30.48 (13.00)	28.38 (8.36)	31.80 (14.45)	20.511	<0.001	0.351	0.319	0.675	0.008

Data are mean (SD). WBVT—Whole body vibration training.

**Table 3 healthcare-09-00652-t003:** Intra-group comparison and post hoc analysis of primary outcome measures.

	WBVT				Control			
	Overall Sig Diff	E1-E2	E1-E3	E2-E3	Overall Sig Diff	E1-E2	E1-E3	E2-E3
Mean Peak Torque	0.002	0.009	0.045	0.857	0.003	0.048	0.012	0.154
Mean Average Power	0.004	0.009	0.033	1.000	<0.001	0.003	0.001	0.280

WBVT—Whole body vibration training; Sig Diff—Significant Difference.

**Table 4 healthcare-09-00652-t004:** Secondary outcome measures (grip strength, SPPB, SF-36 and Body composition analysis).

	E1		E2		E3		Effect Time			Effect Time × Group		
	WBVT	Control	WBVT	Control	WBVT	Control	*F*	*p*	$\eta^{2}p$	*F*	*p*	$\eta^{2}p$
Mean grip strength	26.93 (8.44)	27.05 (9.20)	28.65 (8.88)	27.83 (9.74)	27.93 (8.51)	29.18 (10.08)	3.626	0.038	0.087	1.431	0.246	0.036
SPPB	10.75 (1.12)	10.65 (1.14)	11.40 (0.68)	10.95 (1.00)	11.40 (1.05)	11.25 (0.79)	6.375	0.003	0.144	0.536	0.587	0.014
SF-36	122.65 (14.89)	122.00 (14.98)	124.75 (3.30)	132.30 (10.54)	123.20 (12.55)	131.45 (12.71)	5.483	0.006	0.126	3.104	0.051	0.076
Body weight	59.94 (6.84)	63.06 (14.32)	59.88 (7.02)	62.72 (13.74)	59.90 (6.97)	62.91 (14.00)	0.783	0.442	0.020	0.384	0.647	0.010
Skeletal muscle mass	24.05 (4.12)	24.39 (6.26)	23.71 (4.37)	23.98 (6.08)	23.73 (4.03)	24.09 (5.86)	4.611	0.017	0.108	0.056	0.924	0.001
Body fat mass	15.45 (4.51)	18.23 (8.45)	15.97 (4.56)	18.51 (8.33)	15.98 (4.52)	18.49 (8.57)	2.893	0.067	0.071	0.292	0.748	0.008

Data are mean (SD). WBVT—Whole body vibration training; SPPB—Short Physical Performance Battery; SF-36, 36-Item Short Form Survey.

**Table 5 healthcare-09-00652-t005:** Intra-group comparison and post hoc analysis of secondary outcome measures.

	WBVT				Control			
	Overall Sig Diff	E1-E2	E1-E3	E2-E3	Overall Sig Diff	E1-E2	E1-E3	E2-E3
Mean grip strength	0.094	0.117	0.666	1.000	0.110	1.000	0.254	0.161
SPPB	0.014	0.025	0.058	1.000	0.104	1.000	0.057	0.689
SF-36	0.771	1.000	1.000	1.000	<0.001	0.004	0.006	1.000
Skeletal muscle mass	0.233	0.206	0.712	1.000	0.022	0.007	0.233	1.000

WBVT—Whole body vibration training; Sig Diff—Significant Difference.

## Data Availability

All data generated or analyzed during this study are included in this published article.

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
