# Peer review of "Effectiveness of Whole-Body Vibration Training to Improve Muscle Strength and Physical Performance in Older Adults: Prospective, Single-Blinded, Randomized Controlled Trial"

_healthcare, 2021, doi:10.3390/healthcare9060652_

Round 1

Reviewer 1 Report

I congratulate the authors, as the study is quite well designed and well presented. I only have minor concerns about the explanation of the strengthening exercices protocol, as the exercises are well described in their execution form, but there are no indications about the loads used by patient (for example mean weight used for each exercise compared to main bodyweight of patients). Improving this part could make the study even more clear and complete.

Author Response

I congratulate the authors, as the study is quite well designed and well presented. I only have minor concerns about the explanation of the strengthening exercises protocol, as the exercises are well described in their execution form, but there are no indications about the loads used by patient (for example mean weight used for each exercise compared to main bodyweight of patients). Improving this part could make the study even more clear and complete.

  • Response : The strength exercise was a body weight exercise protocol and there was no additional weight loading. We added the description of loads according to your point. (Page 4, line 3)

Reviewer 2 Report

This article investigated the efficacy of WBVT in improving muscle strength and physical performance in older adults, compared to conventional exercise. As a result, there was significant increase in muscle strength in both group. The increase in mean peak torque was greater in the WBVT group. The SPBB score was significantly increased in WBVT group. Only the control group showed a significant decrease in skeletal muscle mass, and a significant increase in SF-36.

              The Introduction is insufficient to understand background in this study. Please explain more detail or introduce more previous original articles. The explanation in the Method is insufficient to reproduce.

Introduction

  1. P2 Line 4-8 ”WBVT has …”
    The explanation about previous studies in WBVT is not sufficient in the Introduction. Please introduce not only review article, but also some more previous original articles.

  2. P2 L12-14 “Recently in …”
    Authors described in this sentence that the effect of WBVT seem to be small compared with resistance training. If authors would like to clarify this point, you need to set experiment WBVT group vs resistance training group. In this study, both groups include resistance training, thus we can’t separate only the effect of WBVT, stretching or resistance training.

  1. P2 L14-16 “Some point out …”
    Please explain more details about these issues of the previous study. How was insufficient in the previous study? In addition, how do you solve these issues?

  2. P2 L19-21 “The purpose …”
    Authors mentioned that the purpose of this study was to verify the efficacy of WBVT in improving muscle strength. As mentioned Q2, both group performed resistance training in this study. Thus it is difficult to separate the effects of WBVT and resistance training on increasing muscle strength. I think this study would clarify the effect of WBVT + resistance training, or whether WBVT accelerate the effect of resistance training on muscle strength.

    Materials and Methods
  3. P2 L33-34 “The control …”
    Authors performed 20 min stretching. Previous studies reported that acute static stretching decrease muscle strength (e.g. Behm et al. 2011). Is there any adverse effect in resistance exercise performed after stretching in CON group?

  4. P2 L35-39 “Both groups …”
    Participants completed 12 training session. Is this enough for long-term intervention, as authors mentioned in introduction? In addition, is this enough duration to induce muscle adaptation?

  5. P2L44 “2.2. Participants”
    Authors recruited participants at the hospital. Were the participants elderly people who are ill or injured? Also, did participants exercise habitually? Please describe.

  6. P3L8-9 “Third, patients …”
    Were there any patients excluded due to this criteria. What were the specific cases?

  7. P3L16-17 “In this case, …”
    Why did author set device 10Hz for 5min? Please explain.

  8. P3L20-P4L9 “Strengthening exercises …”
    Please add figures or pictures to explain how to perform strength exercises and Stretching.

  9. P4L4-5 “Each exercise …”
    This sentence is not specific for readers. How long did participants perform each exercise? Or, how many times did participants perform each exercise?

  10. P4L7-9 “The stretching exercises …”
    Please explain how to perform stretching.

  11. P4L7-9 “The stretching exercises …”
    Why didn’t author include lower limbs stretching? Strength exercise and vibration training include stimulation to the lower limbs.

  12. P4L9-10 “Each action …”
    How many times or how long did each stretching perform?

  13. P4L15
    Did the author perform measurements for each participant at the same time pre, post and 4 week after post? Especially, the measurement of body composition.

  14. P5L12 “2.4.3. Adverse effects”
    Where is the result about this? How did you evaluate this result objectively?

  15. P5L15 “2.5. Sample size”
    Please move this paragraph before or after the paragraph of participants.

  16. P5L26 “2.6. Randomization”
    Please move this paragraph before or after the paragraph of study design.

  17. P5L34-39 “2.7. Blinding”
    I think this paragraph is not necessary. The sentences “A clinical … investigators.” have already described in the paragraph of study design. Thus, “All participants … assignments” should move to the study design.

  18. P5L40 “2.8. Statistics”
    This statistics confuse readers. In this study, t test should used to compare only ages and K-MMSE between groups. Other values should be analyzed 2-way ANOVAs (time (Pre vs Post vs 4w after post) × group (WBVT vs control)). In addition, authors firstly should have to check significance of interaction. If there is significant difference, you should have to do pairwise comparison. If there is no interaction, you should have to check main effects and, if there is significant difference, pairwise comparison. I can't understand intra-group analysis (Table 3). It is already analyzed by 2-way ANOVAs above mentioned.

  19. P6 L8 ”3. 1. Participants”
    This paragraph and figure2 should move Method section, before or after study design.

  20. P6L14-17 “The mean age …”
    These values have already described in Table. Thus, they need not to describe here again.

  21. P6L15
    What is K-MMSE? Please explain in the methods.

  22. P11L11 “The other …”
    Authors mentioned in the Introduction section that this study include large sample size and long-term intervention. This is discrepancy, compared with this sentence.

  23. P11L14-17 “Lower extremity…”
    This study clarified the effect of WBVT or stretching + strength exercise, thus authors can’t mentioned the effect of only WBVT.

Author Response

Comments and Suggestions for Authors

This article investigated the efficacy of WBVT in improving muscle strength and physical performance in older adults, compared to conventional exercise. As a result, there was significant increase in muscle strength in both group. The increase in mean peak torque was greater in the WBVT group. The SPBB score was significantly increased in WBVT group. Only the control group showed a significant decrease in skeletal muscle mass, and a significant increase in SF-36.

The Introduction is insufficient to understand background in this study. Please explain more detail or introduce more previous original articles. The explanation in the Method is insufficient to reproduce.

Introduction

1.P2 Line 4-8 ”WBVT has …”

The explanation about previous studies in WBVT is not sufficient in the Introduction. Please introduce not only review article, but also some more previous original articles.

  • Response : We have added a description of recent studies on WBVT in your opinion. (Page 2, line 4-12)

2.P2 L12-14 “Recently in …”

Authors described in this sentence that the effect of WBVT seem to be small compared with resistance training. If authors would like to clarify this point, you need to set experiment WBVT group vs resistance training group. In this study, both groups include resistance training, thus we can’t separate only the effect of WBVT, stretching or resistance training.

  • Response : We also agreed to the consideration mentioned in your question 4 and reoriented the introduction section. We focused on the analysis of the effect of WBVT + strengthening. (Page 2, line 15-23)

3.P2 L14-16 “Some point out …”

Please explain more details about these issues of the previous study. How was insufficient in the previous study? In addition, how do you solve these issues?

  • Response : This description has been deleted.

4.P2 L19-21 “The purpose …”

Authors mentioned that the purpose of this study was to verify the efficacy of WBVT in improving muscle strength. As mentioned Q2, both group performed resistance training in this study. Thus it is difficult to separate the effects of WBVT and resistance training on increasing muscle strength. I think this study would clarify the effect of WBVT + resistance training, or whether WBVT accelerate the effect of resistance training on muscle strength.

  • Response : As mentioned in answer 2, the introduction was revised according to your opinion. (Page 2, line 15-23)

Materials and Methods

5.P2 L33-34 “The control …”

Authors performed 20 min stretching. Previous studies reported that acute static stretching decrease muscle strength (e.g. Behm et al. 2011). Is there any adverse effect in resistance exercise performed after stretching in CON group?

  • Response : As you mentioned, acute static stretching reduces muscle strength. This negative effect is known to be especially affected when stretching is performed immediately prior to the muscle strength activity and when doing activities that require high power. (https://doi.org/10.1111/j.1600-0838.2012.01444.x, https://doi.org/10.1111/j.1600-0838.2009.01058.x) But, in our study, it was dynamic or slow movement stretching and performed with relatively low intensity (point of mild discomfort), and strengthening exercise (body weight exercise, not strenuous exercise) was performed after 10 minutes of rest. In our protocol, the decrease in muscle strength by stretching was considered to be minimal, and there were no patient complaints or adverse effect. (Page 4, line 23 ~ Page 5, line 3)

6.P2 L35-39 “Both groups …”

Participants completed 12 training session. Is this enough for long-term intervention, as authors mentioned in introduction? In addition, is this enough duration to induce muscle adaptation?

  • Response : Although the training protocols of previous studies have varied, we set 12 sessions (3 sessions per week, 4 weeks) by referring to studies that showed significant results about muscle strength in the 3-4 weeks (less than 12 sessions) protocol. (https://doi.org/10.1177/1352458511423267, https://doi.org/10.3109/09593980902967196)

7.P2L44 “2.2. Participants”

Authors recruited participants at the hospital. Were the participants elderly people who are ill or injured? Also, did participants exercise habitually? Please describe.

-          Response : Our subjects were elderly with no major disease conditions including musculoskeletal injury, presented in the inclusion/exclusion criteria. They didn't exercise regularly other than intermittent walking exercises, but they were relatively healthy elderly people with no problems on their daily activities.

8.P3L8-9 “Third, patients …”

Were there any patients excluded due to this criteria. What were the specific cases?

  • Response : One patient was excluded due to cardiovascular disease (vascular thrombosis). No subjects were excluded by the third criteria: subjects deemed to be ineligible for participation.

9.P3L16-17 “In this case, …”

Why did author set device 10Hz for 5min? Please explain.

  • Response : In previous studies, the frequency of the vibration signals used ranged from 10 to 54 Hz, with an amplitude between 0.05 mm and 8 mm. We set the frequency and amplitude as 10Hz and 5mm as the most effective values in previous studies using similar equipment (https://doi.org/10.5103/KJSB.2015.25.3.343, https://doi.org/10.5103/KJSB.2015.25.4.383).

10 & 12.

10.P3L20-P4L9 “Strengthening exercises …”

Please add figures or pictures to explain how to perform strength exercises and Stretching.

12.P4L7-9 “The stretching exercises …”

Please explain how to perform stretching.

  • Response : We have prepared a training program by referring to The American College of Sports Medicine (ACSM) Guidelines (10.1249/MSS.0b013e3181915670 , 10.1249/01.MSS.0000142662.21767.58, ACSM's Complete Guide to Fitness & Health, 2nd Edition, ACSM's Resources for the Certified Personal Trainer, 5th Edition, ACSM Exercise Photos (google.com). Figures are presented in the guideline book. References have been added for the action and will be replaced with the description of the manuscript. (Page 4, line 3-23)

  1. & 14.
  2. P4L4-5 “Each exercise …”

This sentence is not specific for readers. How long did participants perform each exercise? Or, how many times did participants perform each exercise?

14.P4L9-10 “Each action …”

How many times or how long did each stretching perform?

  • Response : We prepared a total of 10 stretching motion. Each motion was performed slowly over 4 seconds per repetition. Subjects repeated each motion for 30 seconds and then repeated with the next motion. It took a total of 5 minutes to repeat all 10 stretching for 30 seconds, and this was repeated 4 times for 20 minutes (4 sets x (10 motion x 30s )). This description has been added to the manuscript. (Page 4, line20-23)

13.P4L7-9 “The stretching exercises …”

Why didn’t author include lower limbs stretching? Strength exercise and vibration training include stimulation to the lower limbs.

  • Response : Each specific motion has been described in detail. (Page 4, line 14-23)

15.P4L15

Did the author perform measurements for each participant at the same time pre, post and 4 week after post? Especially, the measurement of body composition.

  • Response : Two blinded evaluators who did non participate in intervention conducted the evaluation, including of body composition analysis. For one subject, the same evaluator performed three repeated measurements (pre, post and 4 weeks after). (Page 5, line 10-11)

16.P5L12 “2.4.3. Adverse effects”

Where is the result about this? How did you evaluate this result objectively?

  • Response : There were no significant adverse effects and this was added to the end of the results section. (Page 10, line 21) During the evaluation, vital signs were measured, and the participant was able to report all adverse effects including subjective symptoms to the evaluator. (Page 6, line 6-8) However, there was no significant adverse effect.

17.P5L15 “2.5. Sample size”

Please move this paragraph before or after the paragraph of participants.

  • Response : The paragraph of sample size was moved after the paragraph of participants. (Page 3, line 22)

18.P5L26 “2.6. Randomization”

Please move this paragraph before or after the paragraph of study design.

  • Response : The paragraph of randomization was moved after the paragraph of participants. (Page 2, line 48)

19.P5L34-39 “2.7. Blinding”

I think this paragraph is not necessary. The sentences “A clinical … investigators.” have already described in the paragraph of study design. Thus, “All participants … assignments” should move to the study design.

  • Response : “All participants … assignments” was moved to the study design. (Page 2, line 32-33) The paragraph of blinding was deleted.

20.P5L40 “2.8. Statistics”

This statistics confuse readers. In this study, t test should used to compare only ages and K-MMSE between groups. Other values should be analyzed 2-way ANOVAs (time (Pre vs Post vs 4w after post) × group (WBVT vs control)). In addition, authors firstly should have to check significance of interaction. If there is significant difference, you should have to do pairwise comparison. If there is no interaction, you should have to check main effects and, if there is significant difference, pairwise comparison. I can't understand intra-group analysis (Table 3). It is already analyzed by 2-way ANOVAs above mentioned.

  • Response : Non-demographic outcome variables were excluded from Table 1.

This study was repeated the measurements three times in two groups. Repeated measurements within the same group could not be analyzed with 2-way ANOVA because the results were not independent, and had no choice but to be analyzed with RM-ANOVA.

As mentioned in the manuscript, the group effects and time x group interactions were not significant. In the first RM-ANOVA, the results of time are derived regardless of the group, so we performed post-hoc analyses to know the time effect of each group.

21.P6 L8 ”3. 1. Participants”

This paragraph and figure2 should move Method section, before or after study design.

  • We reported according to the CONSORT guidelines, and according to it, the results section should contain information on participants (the numbers, assignments, and received treatments) including participant flows and diagram.

22.P6L14-17 “The mean age …”

These values have already described in Table. Thus, they need not to describe here again.

  • Response : According to your opinion, the repeated description already presented in the table 1 has been deleted from the text.

23.P6L15

What is K-MMSE? Please explain in the methods.

  • Response : K-MMSE is a Korean version of Mini-Mental State Examination, and a full term has been added to the method. (Page3, line 15)

24.P11L11 “The other …”

Authors mentioned in the Introduction section that this study include large sample size and long-term intervention. This is discrepancy, compared with this sentence.

  • Response : That phrase that conflicts with our limitation has been deleted.

25.P11L14-17 “Lower extremity…”

This study clarified the effect of WBVT or stretching + strength exercise, thus authors can’t mentioned the effect of only WBVT.

-          Response : We agree with you. The description has been revised so as not to be misunderstood like the effect of only WBVT. (Page 12, line 7-11)

Round 2

Reviewer 2 Report

Revise2-1
5.P2 L33-34 “The control …”
Authors performed 20 min stretching. Previous studies reported that acute static stretching decrease muscle strength (e.g. Behm et al. 2011). Is there any adverse effect in resistance exercise performed after stretching in CON group?
Response : As you mentioned, acute static stretching reduces muscle strength. This negative effect is known to be especially affected when stretching is performed immediately prior to the muscle strength activity and when doing activities that require high power. (https://doi.org/10.1111/j.1600-0838.2012.01444.x, https://doi.org/10.1111/j.1600-0838.2009.01058.x) But, in our study, it was dynamic or slow movement stretching and performed with relatively low intensity (point of mild discomfort), and strengthening exercise (body weight exercise, not strenuous exercise) was performed after 10 minutes of rest. In our protocol, the decrease in muscle strength by stretching was considered to be minimal, and there were no patient complaints or adverse effect. (Page 4, line 23 ~ Page 5, line 3)

Q. I understand your opinion. This explanation prefer to be included in the Discussion section.

Revise2-2
13.P4L7-9 “The stretching exercises …”
Why didn’t author include lower limbs stretching? Strength exercise and vibration training include stimulation to the lower limbs.
Response : Each specific motion has been described in detail. (Page 4, line 14-23)

Q. I understand how participants performed stretching exercise. But, can we be called this exercise as stretching? What type of stretching does this exercise be categorized (static, dynamic or ballistic)? Dynamic stretching is a bit firster than this exercise (for ex. Samukawa et al. 2011). To prevent confusion for readers, I suggest this exercise would prefer to be described as “slow movement exercise”.

Revise2-3
15.P4L15
Did the author perform measurements for each participant at the same time pre, post and 4 week after post? Especially, the measurement of body composition.
Response : Two blinded evaluators who did non participate in intervention conducted the evaluation, including of body composition analysis. For one subject, the same evaluator performed three repeated measurements (pre, post and 4 weeks after). (Page 5, line 10-11)

Q. Author misunderstand my question. I want to know whether each participant was assessed at same time zone among pre, post or 4 weeks after. For ex. every morning, at 10 AM, etc. I’m afraid that if the measurement time was different within individual (for ex. pre was massessed at 10am, post at 3pm ,4 weekd after at 8pm), diurnal fluctuation would be affect the results, especially body composition.

Author Response

Comments and Suggestions for Authors

Revise2-1

5.P2 L33-34 “The control …”

Authors performed 20 min stretching. Previous studies reported that acute static stretching decrease muscle strength (e.g. Behm et al. 2011). Is there any adverse effect in resistance exercise performed after stretching in CON group?

Response : As you mentioned, acute static stretching reduces muscle strength. This negative effect is known to be especially affected when stretching is performed immediately prior to the muscle strength activity and when doing activities that require high power. (https://doi.org/10.1111/j.1600-0838.2012.01444.x, https://doi.org/10.1111/j.1600-0838.2009.01058.x) But, in our study, it was dynamic or slow movement stretching and performed with relatively low intensity (point of mild discomfort), and strengthening exercise (body weight exercise, not strenuous exercise) was performed after 10 minutes of rest. In our protocol, the decrease in muscle strength by stretching was considered to be minimal, and there were no patient complaints or adverse effect. (Page 4, line 23 ~ Page 5, line 3)

  1. I understand your opinion. This explanation prefer to be included in the Discussion section.

- Response : That explanation has been moved to the discussion section. (Page 10, line 44~page 11, line 4)

Revise2-2

  1. I understand how participants performed stretching exercise. But, can we be called this exercise as stretching? What type of stretching does this exercise be categorized (static, dynamic or ballistic)? Dynamic stretching is a bit firster than this exercise (for ex. Samukawa et al. 2011). To prevent confusion for readers, I suggest this exercise would prefer to be described as “slow movement exercise”.

- Response : We agree with your point. It was revised as “slow movement exercise”. (Page 4, line 13~14)

Revise2-3

15.P4L15

  1. Author misunderstand my question. I want to know whether each participant was assessed at same time zone among pre, post or 4 weeks after. For ex. every morning, at 10 AM, etc. I’m afraid that if the measurement time was different within individual (for ex. pre was assessed at 10am, post at 3pm ,4 week after at 8pm), diurnal fluctuation would be affect the results, especially body composition.

- Response : We didn't understand the question properly at first. All outcome measurements, including body composition analysis, were taken in the morning (A.M.) time zone. We have added a description about your question to the methods. (Page 5, line 50~51)
